# Effects of Resistance Exercise and Essential Amino Acid Intake on Muscle Quality, Myokine, and Inflammation Factors in Young Adult Males

**DOI:** 10.3390/nu16111688

**Published:** 2024-05-29

**Authors:** Deokhwa Jeong, Kyumin Park, Jinseok Lee, Jiye Choi, Haifeng Du, Hyeongmo Jeong, Liangliang Li, Kenji Sakai, Sunghwun Kang

**Affiliations:** 1Department of Smart Health Science and Technology, Kangwon National University, Chuncheon 24341, Gangwon-do, Republic of Korea; 93deokgoo@kangwon.ac.kr (D.J.); cjyeaaaah@gamil.com (J.C.); dhf2051711464@gmail.com (H.D.); 2Center for Sports Science in Gangwon, Chuncheon 24239, Gangwon-do, Republic of Korea; katckyumin@gmail.com; 3Department of Sport Science, Kangwon National University, Chuncheon 24341, Gangwon-do, Republic of Korea; jinseok0515@gmail.com (J.L.); hyeongmo2342@naver.com (H.J.); liliangliang@kangwon.ac.kr (L.L.); 4Chemicals & Life Science Division, Nagase Korea Corporation, Seoul 04527, Republic of Korea; kenji.skai@nagase.co.jp

**Keywords:** resistance exercise, essential amino acids, muscle quality, myokines, inflammation

## Abstract

Background: Recently, many studies have been devoted to discovering nutrients for exercise-like effects. Resistance exercise and the intake of essential amino acids (EAAs) are known to be factors that can affect muscle mass and strength improvement. The purpose of this study was to investigate changes in muscle quality, myokines, and inflammation in response to resistance exercise and EAA supplementation. Methods: Thirty-four males volunteered to participate in this study. They were assigned to four groups: (1) placebo (CO), (2) resistance exercise (RE), (3) EAA supplementation, and (4) RE + EAA supplementation. Body composition, muscle quality, myokines, and inflammation were measured at baseline and four weeks after treatment. Results: Lean body fat had decreased in both RE and RE + EAA groups. Lean body mass had increased in only the RE + EAA group. In all groups except for CO, irisin, myostatin A, and TNF-α levels had decreased. The grip strength of the right hand and trunk flexion peak torque increased in the RE group. The grip strength of the left hand, trunk flexion peak torque, and knee flexion peak torque of the left leg were increased in RE + EAA. Conclusions: RE, EAA, and RE + EAA could effectively improve the muscle quality, myokine, and inflammation factors of young adult males. This finding highlights the importance of resistance exercise and amino acid intake.

## 1. Introduction

Physical inactivity is considered one of the reasons for the development of chronic diseases. Its rate has been gradually increasing since COVID-19 [1]. Reduced physical activity can cause negative changes in body composition, such as fat accumulation and muscle atrophy. These negative changes in body composition are known to contribute to approximately 6–10% of global mortality, with a negative impact on non-communicable diseases such as cardiovascular disease, obesity, and diabetes [2,3]. Conversely, consuming excess energy from body fat and increasing muscle mass through regular physical activity can increase life expectancy by preventing metabolic disease, cardiopulmonary diseases, and chronic diseases, thus reducing mortality [4,5,6].

During resistance exercise, skeletal muscle can promote protein metabolism in response to exercise stimuli. In particular, muscle protein synthesis is increased for up to 24 h after resistance exercise, contributing to muscle hypertrophy [7]. During resistance exercise, the secretion of myokines and cytokines is promoted in the muscles [8,9]. These secreted myokines can promote energy consumption by activating metabolism throughout the body, thus positively improving cardiovascular factors, neuroplasticity, and chronic inflammation [10]. Furthermore, stimulating the secretion of myokines is directly correlated with the improvement of physical inactivity [11,12,13,14,15]. Therefore, increasing muscle mass and improving the secretion of myokines through resistance exercise are essential elements for improving health.

Amino acids are the basic building blocks of proteins in the body. They must be constantly replenished for protein and muscle synthesis [16]. Essential amino acids (EAAs) cannot be synthesized in the body. Thus, they must be sourced from outside it. These EAAs are essential for muscle protein synthesis (MPS) to increase the concentration of EAAs in the body, thereby reducing muscle protein breakdown (MPB). They can increase muscle mass and muscle quality [17]. Among them, leucine plays an important role in regulating protein synthesis through direct action on the mammalian target of rapamycin (mTOR) complex 1 (mTORC1) signaling pathway, which can activate MPS [18]. Additionally, because it has a positive effect on mitochondrial synthesis [19] and energy metabolism in the body, the continuous intake of leucine-rich EAAs is an important strategy for improving muscle mass and muscle quality.

Many clinical trials have reported the positive effects of EAA supplementation in improving protein synthesis, glucose metabolism, and activating MPS [19,20,21,22,23,24]. However, these studies were medium/long-term studies. They could not verify short-term (4-week) effects. Additionally, studies on myokine changes following EAA intake are insufficient. Since the intake of EAAs could improve muscle quality, this is expected to have a positive effect on myokines. Therefore, research is needed on the effects of short-term (4-week) EAA supplementation and resistance exercise on muscle mass, muscle quality improvement, myokines, and inflammation factors. Accordingly, this study aimed to test the following hypotheses: (1) EAA intake could have a positive effect on the improvement of inflammation factors, myokine secretion, and muscle quality, similar to resistance exercise; (2) combining EAA intake with resistance exercise could have greater effects on muscle mass, muscle quality, myokines, and inflammation factors than EAA intake or resistance exercise alone.

## 2. Methods

### 2.1. Study Design

We conducted this randomized controlled trial in order to examine the effects of various factors on the control group (CON), resistance exercise group (RE), EAA intake group (EAA), and the resistance exercise + EAA intake group (RE + EAA). The participants, excluding those who dropped out, were divided into four groups: CON (*n* = 8), RE (*n* = 10), EAA (*n* = 9), and RE + EAA (*n* = 7). Eligible participants attended the exercise physiology laboratory at the Department of Sports Science at baseline (BL) and 4 weeks after the experimental intervention (POST), with each assessment following an overnight fast. The laboratory visits included a body composition assessment using a body composition analyzer. Also, to evaluate muscle strength (kg), a digital grip dynamometer (GRIP-D 5101; TAKEI, Co., Tokyo, Japan) was used to measure grip strength, and trunk and knee strength tests were performed at an angular velocity of 60°/s, followed by extension and flexion strength tests recorded three times using an isokinetic dynamometer machine (Hu-mac Norm Testing and Rehabilitation, CSMi Medical & Solution, Stoughton, MA, USA). A venous blood sample was drawn from an antecubital vein to assess myokine and inflammation factors levels. During week 1 of the intervention, all sessions were supervised by an exercise physiologist, whilst only one session per week was supervised during weeks 2 to 4, with overall supervision of the experimental intervention. Participants were asked to keep a weekly exercise and EAA intake diary. Specific contents of the study design are shown in Figure 1.

### 2.2. Participation

This study complied with the ethical standards of the Declaration of Helsinki. It was approved by the Institutional Review Board (IRB) of Kangwon National University for Human Subjects (KWNUIRB-2022-08-006-002). Forty male college students voluntarily participated in this study from 8 September 2022 to 31 January 2023, after obtaining IRB approval.

Inclusion criteria were: (1) those who could perform resistance exercise, and (2) those who had no allergic reaction to amino acid intake. Exclusion criteria were: (1) those who were taking other types of protein or amino acid supplements, (2) those who experienced cardiovascular or cardiopulmonary dysfunction, and (3) those who had experienced musculoskeletal injuries over the past six months. The sample size was calculated using G* Power software, version 3.1.9.7 (Heinrich Heine University, Düsseldorf, Germany). By setting α = 0.05 and β = 0.80, the sample size was calculated to be 34 people. Allowing for possible dropout, 40 people were recruited. All subjects participated after completing a consent form. They were randomly grouped by selecting a piece of paper from an opaque box with A, B, C, and D written on it: A, placebo/control (CO), *n* = 10; B, resistance exercise (RE), *n* = 10; C, essential amino acid supplements (EAA), *n* = 10; and D, resistance exercise combined with essential amino acid supplements (RE + EAA), *n* = 10. During participation, 6 participants (CO, *n* = 2; EAA, *n* = 1; and RE + EAA, *n* = 3) withdrew from the experiment. During the experiment period, excluding weekends, the average dietary calories consumed were 2738 ± 124 kcal (carbohydrate: 1372 ± 260 kcal; protein: 492 ± 116 kcal; fat: 783 ± 171 kcal). The participants’ characteristics are shown in Table 1.

### 2.3. Measurement of Body Composition

The body composition variables of the participants were measured. Using a body composition analyzer (Inbody 720 Body Composition Analyzer, Biospace, Seoul, Republic of Korea), weight (kg), lean body fat (kg), and lean body mass (kg) were measured to the nearest 0.1 kg. Body mass index (BMI) was determined by dividing each participant’s weight in kilograms by the square of their height in meters.

### 2.4. Measurement of Muscle Quality (Grip, Trunk, and Knee Strength Test)

To evaluate muscle strength (kg), a digital grip dynamometer (GRIP-D 5101; TAKEI, Co., Tokyo, Japan) was used to measure grip strength. All measurements were entered into an electronic card and transmitted to a computer [25]. Trunk and knee strength tests were performed at an angular velocity of 60°/s, followed by extension and flexion strength tests performed three times using an isokinetic dynamometer machine (Humac Norm Testing and Rehabilitation, CSMi Medical & Solution, Stoughton, MA, USA). During the measurement of knee strength, the lateral epicondyle of the femur and the axis of the Cybex were aligned, and the whole body was fixed with a belt so that no force other than from the knee joint was applied. The subject’s position was raised to adjust the posture. The height of the footrest and the scapular pad were adjusted to suit the subject’s height to measure trunk strength. Angles of flexion and extensor muscles were set by fixing the upper body and the lower body and holding the handle of the chest pad with both hands. After sufficient practice before the test, this measurement was conducted once the motion was familiar. Analysis variables were maximum flexion and extension peak torque per body weight (Newton-meter/body weight) [26].

### 2.5. Hematological Analysis

Fasting venous blood samples were collected from all participants at baseline and 4 weeks after treatment. Fasting was maintained for 12 h after eating a meal on the previous day. Blood samples were collected on the following day, after adequate sleep and refraining from radical movement as much as possible. All samples were drawn at 08:00 from antecubital veins. The collected blood samples were immediately centrifuged at 3500 g for 10 min at room temperature. Serum samples were stored at −80 °C until further analysis [22]. Serum levels of myokine biomarkers such as irisin, myostatin, and cathepsin B (DY9420-05, DGDF80, and DY2176: R&D Systems, Minneapolis, MN, USA), and inflammation biomarkers such as IL-6 and tumor necrosis factor-alpha (TNF-a) (R6000B and DTA00D: R&D Systems, Minneapolis, MN, USA) were measured using DuoSet ^TM^ enzyme-linked immunosorbent assay (ELISA) kits (R&D Systems, Minneapolis, MN, USA) according to the manufacturer’s instructions.

### 2.6. Essential Amino Acids Supplement

Subjects in the EAA and RE + EAA groups were supplemented with essential amino acid products (EAALPHA, Prinova Group LLC, Itasca, IL, USA) twice daily for 4 weeks. The EAA intake time was within 30 min after a workout or between meals. Participants were instructed to take one packet of EAAs with pure water once a day for 4 weeks. The intake time was set after the end of exercise on days when the subject was participating in the exercise program. The intake was allowed freely on days when the subject was not participating in the exercise program. For 4 weeks, the participants were supervised to ensure that they were consuming the EAAs properly. The components of the EAAs are described in Table 2.

### 2.7. Resistance Exercise Program

All participants except those in the CO and EAA groups were familiar with resistance exercises. They proceeded with an exercise program lasting for a total of 70 min, including 10 min of warm-up exercise, 50 min of main exercise, and 10 min of cool-down exercise. The resistance exercise protocol, modified from our previous study [26], consisted of 9 items with 3 sets per exercise item, conducted 4 times a week for 4 weeks. The exercise intensity was 1RM 60~70% for the first week and 1RM 70~80% for weeks 2 to 4, according to the purposes of muscle hypertrophy [27,28]. The exercise volume was set to consume 300 to 350 kcal per day. Specific contents of the exercise program are shown in Table 3.

### 2.8. Statistical Analysis

All results are reported as mean ± standard deviation. All data were analyzed using SPSS version 25.0 (SPSS Inc., Chicago, IL, USA). First, a two-way repeated analysis of variance (ANOVA) was used to assess the main effect in terms of group and time and the interaction effect between group and time in variables. Second, the main effect was analyzed using a one-way ANOVA for the between-group effect and paired-sample *t*-tests for the within-group effect. Statistical significance was accepted at α = 0.05. Statistically nonsignificant trends are only expressed by up to 0.1 in the text.

## 3. Results

### 3.1. Changes in Body Composition

Changes in body compositions are shown in Table 4. The two-way repeated ANOVA showed a grouping by time interaction effects on lean body fat (*p* = 0.025) and lean body mass (*p* = 0.010). BMI (*p* = 0.034), lean body fat (*p* = 0.009), and lean body mass (*p* = 0.019) showed significant main effects with respect to time. The difference according to time was identified through paired *t*-tests. RE (*p* = 0.020) and RE + EAA (*p* = 0.037) values at four weeks showed significantly decreased lean body fat compared to baseline. However, lean body mass was only significantly increased with RE + EAA (*p* = 0.033) for four weeks compared to baseline.

### 3.2. Changes in Muscle Quality

Changes in muscle quality are shown in Table 5. The two-way repeated ANOVA showed a group-by-time interaction effect on trunk flexion (*p* = 0.048). Grip strength (L) (*p* = 0.044), grip strength (R) (*p* = 0.003), trunk flexion peak torque (*p* = 0.05), and knee flexion peak torque (L) (*p* = 0.040) showed significant main effects with respect to time. The difference according to time was identified through paired *t*-tests. After four weeks, the grip strength of the left hand in the RE + EAA group (*p* = 0.019) had significantly increased compared to baseline. In addition, the grip strength of the right hand in the RE (*p* = 0.006) had significantly increased compared to the baseline. Changes in trunk strength showed a trending main effect of time after treatment (*p* = 0.05). The flexion peak torque of the left knee tended to increase compared to the baseline (*p* = 0.05). The trunk flexion in the RE (*p* = 0.000) had significantly increased compared to the baseline. In addition, trunk flexion had significantly increased in the RE + EAA group (*p* = 0.006) compared to baseline. Changes in knee strength showed a trending main effect of time. The knee flexion in the RE + EAA group was significantly increased compared to baseline (*p* = 0.05).

### 3.3. Changes in Myokine and Inflammation Factors

Changes in myokine and inflammation factors are shown in Figure 2. There was a significant interaction effect on TNF-α (*p* = 0.000), with a tendency of interaction effect on myostatin A (*p* = 0.058). There were no main effects between groups on any variables.

The myokine factors, irisin (*p* = 0.000) and myostatin A (*p* = 0.000), and cathepsin B (*p* = 0.015) showed significant main effects over time. Also, the inflammation factor TNF-α (*p* = 0.000) showed significant main effects over time. The myokine factor irisin levels were significantly increased in the RE (*p* = 0.006), EAA (*p* = 0.008), and RE + EAA (*p* = 0.003) groups at four weeks compared to baseline. Myostatin A levels were significantly decreased in RE (*p* = 0.000), EAA (*p* = 0.001), and RE + EAA (*p* = 0.007) groups at four weeks compared to baseline. Cathepsin B tended to have increased in the RE + EAA groups compared to baseline. The inflammation factor TNF-α levels had also significantly decreased in the RE (*p* = 0.002), EAA (*p* = 0.000), and RE + EAA (*p* = 0.001) groups after four weeks of treatment compared to baseline.

## 4. Discussion

This study analyzed the effects of four weeks of resistance exercise (RE) and EAA treatment on muscle mass, muscle quality, myokine, and inflammation factors in men in their 20s. The following results were obtained: (1) lean body mass only increased with the combined treatment of RE and EAA; (2) RE and EAA treatment alone or in combination showed positive changes in myokines (increase in irisin, decrease in myostatin A); (3) inflammation factor (TNF-α) levels decreased after treatment with RE and EAA alone or in combination; (4) muscle quality was improved in the RE group (grip strength—right) and the RE + EAA group (grip strength—left, trunk flexion peak torque, and knee flexion peak torque—right).

### 4.1. How Do Resistance Exercise and EAA Intake Treatment Affect Muscle Mass?

Resistance exercise and dietary protein intake are important strategies for stimulating MPS, promoting muscle hypertrophy, and increasing muscle mass. Increased muscle mass occurs when MPS exceeds MPB, and the net protein balance is high [29]. Several clinical studies have shown that regular RE and EAA supplementation, both alone and in combination, can increase muscle mass through the activation of MPS and the reduction of MPB [30,31,32]. However, according to a meta-analysis [33] conducted to verify the clinical effects of RE and protein intake, the duration of the RE program was set at 6 to 24 weeks (mean ± SD: 12 weeks ± 5 weeks), so it was unclear whether short-term (< 5 weeks) RE and EAA treatment was effective in increasing muscle mass. Therefore, this study set the duration of treatment to be a short period to determine changes in body composition (increased muscle mass) when a combined treatment involving RE and EAA supplementation was performed. In this study, a significant increase (*p* < 0.05) in muscle mass was observed in the RE + EAA group, whereas no change in muscle mass was found in the RE or EAA single-treatment group. These results were in line with the study by Dillon et al., which showed that treatment with RE + EAA could activate intramuscular amino acid transporters more effectively than RE alone in young adults [34]. In animal experiments, compared to 4 weeks of myostatin inhibitor (ACV) treatment alone, an ACV + RE + EAA combined treatment significantly increased muscle mass and strength [35], which is consistent with the results of the present study. However, in contrast to our study, Antonio et al. reported that when untrained adult women were treated with RE and EAA (18.3 g/day) for 6 weeks, no changes in fat or muscle mass were found [36]. These results show that even within similar age groups, the degree of MPS can vary depending on the subject’s gender and hormonal influences. According to previous research, the acute ingestion of EAA resulted in a diminished accretion of MPS in older compared with younger adults, despite similar increases in blood EAA concentrations [37]. Thus, it is necessary to confirm the most effective duration of RE + EAA considering the subject’s exercise intensity, gender, and age. In addition, although RE or EAA treatment alone can activate MPS for a short period of time, it might be insufficient to increase muscle mass.

### 4.2. What Effects Do Resistance Exercise and EAA Treatment Have on Muscle Quality?

Muscle mass is a major factor in determining muscle strength [38] and the main target of protein. It is an organ that is more affected by changes in protein intake than other tissues. The dietary intake of EAAs is known to stimulate muscle synthesis by activating the mammalian/mechanistic target of the rapamycin complex 1 pathway [39]. In particular, the leucine in EAAs can independently activate mTORC1 and downstream anabolic signaling [40] and bind to Setrin-2, the intracellular “LEU sensor” of the mTORC1 upstream pathway, to separate GTPase-activating protein (GATOR2), thereby activating mTORC1. It plays a role in activating the path. Ikeda et al. reported that RE and amino acid intake for four weeks had a significant effect on knee extension strength [23]. Additionally, Hoffman et al. found a 22% greater change in the delta 1-RM of squats in strength/power athletes with protein consumption than those without protein consumption [41]. Our study also found significant changes in left grip strength (*p* < 0.05), trunk flexion peak torque (*p* < 0.05), and knee flexion peak torque R (*p* < 0.05) with the increase in muscle mass in the RE + EAA group. Interestingly, despite a short-term treatment (RE + EAA) for four weeks, the adaptive response of muscle nerves, which is a representative response to an increase in muscle strength [42], can encourage the upregulation of mitochondrial protein turnover and increase the protein synthesis rate [35], and appear to increase the lean body mass and muscle quality factors. Some factors of quality also appeared to increase. These results partially verified our second hypothesis. It appears that dietary EAA plays a role in improving muscle mass and muscle quality. Thus, a combined treatment of resistance exercise and EAA is believed to have potential benefits in generating muscle mass and improving muscle quality, not only in people with obesity, the elderly, and with sarcopenia but also in healthy adults. Conversely, the first hypothesis that muscle mass and muscle quality could be improved by EAA intake alone was not proven. This was not consistent with a previous study showing that EAA supplementation could induce strength improvement without changing muscle mass [43]. These results indicated that treatment with EAA alone for four weeks (short-term) did not upregulate mitochondrial protein turnover or increase the protein synthesis rate. In addition, there is evidence that long-term EAA supplementation alone can lead to muscle strength improvement without changing muscle mass [43]. However, resistance exercise treatment is considered essential to improving muscle strength following short-term treatment.

### 4.3. How Do Resistance Exercise, EAA, and Their Combination Affect Myokine and Inflammation Factors?

Muscle hypertrophy occurs as a result of skeletal muscle contractions through chemical messengers called myokines or exerkines. Currently, more than 600 myokines have been discovered that respond to skeletal muscle contraction. The most studied myokines include IL-6, IL-15, myostatin, irisin, BDNF, FGF-21, follistatin, and metrnl [44,45]. Myokines can induce protein synthesis (myostatin has the opposite effect) by activating the Akt/mTOR pathway [46]. Additionally, it is involved in muscle hypertrophy and muscle recovery through insulin sensitivity, fat oxidation, muscle formation, and mitochondrial production [47,48].

We confirmed that irisin significantly (*p* < 0.05) increased, while myostatin A significantly (*p* < 0.05) decreased in all groups (RE, EAA, RE + EAA). These results support the results of previous research showing that RE and EAA intake can promote muscle hypertrophy and synthesis by activating PGC-1α and increasing the expression of irisin [46]. In addition, an increase in irisin in the body through skeletal muscle contraction is known to be effective in reducing body fat through the browning of white fat (WAT) by activating UCP-1 [49]. Although a significant decrease in body fat was not confirmed in this study, an increase in irisin was confirmed. Thus, if the treatment period for RE and EAA is set to be more than four weeks or if the study subjects are classified as obese, it has the potential to have a positive effect on body fat reduction. The basis for this was confirmed.

Myostatin can bind to the activin receptor IIB (ActRIIB), activating Smad2/3 and inhibiting the expression of MyoD, which are known to play an important role in muscle synthesis and recovery [50]. Additionally, the myostatin-induced translocation of FoxOs to the nucleus can inhibit Akt1 and accelerate protein degradation [51]. The overexpression of myostatin can induce muscle wastage by suppressing Akt1/mTORC1 signaling in vivo and reducing the myotube diameter through the increase of atrogin-1 and the suppression of MyoD expression [52]. According to previous studies, RE and EAA treatment (either alone or in combination) can downregulate myostatin levels [53], which is positive for muscle synthesis through the activation of Akt1/mTORC1 signaling [54]. Our study was similar to previous studies showing the downregulation of myostatin following the single or combined treatment of RE and EAA [55]. Additionally, preliminary evidence confirmed the activation of Akt1/mTORC1 signaling and muscle synthesis through a reduction in myostatin. Meanwhile, TNF-α is an inflammation cytokine. Disease or its overexpression is known to induce muscle atrophy [46]. Conversely, TNF-α is an inflammation cytokine, and disease or overexpression is known to induce muscle atrophy [46]. Accordingly, the overexpression of TNF-α appears to be a negative factor in muscle hypertrophy and recovery. We confirmed a significant decrease in TNF-α in all treated groups (RE, EAA, and RE + EAA). These results were consistent with previous studies showing that the irisin produced by muscle contraction reduced inflammation expression, not only in macrophages but also in epithelial cells [56], and that the expression of mTOR could suppress inflammation expression [57]. RE and EAA intake are effective in activating irisin and inhibiting the activation of myostatin A, thereby suppressing the expression of mTOR and inflammation cytokines. Therefore, we found preliminary evidence that the improvement of myokine levels and inflammation factors could improve MPS. In particular, RE + EAA treatment is most likely to increase MPS and suppress MPB. Thus, RE + EAA treatment could affect muscle mass improvement factors in the short term.

This study has several potential limitations. First, the sample size was small. Future studies with larger sample sizes should characterize the effects of EAA+RE on myokine, inflammation factors, and muscle quality in young adults. Second, although we restricted average calorie intake except at the weekend, we could not control habitual activity levels. Future studies should consider controlling the amount of physical activity during the experiment.

Third, when dividing the subjects of this study into groups, they were divided randomly using a placement method for fairness. However, body weight, BMI, lean body mass, grip strength, and trunk strength did not start at the same level at the beginning of the study. Lastly, muscle protein metabolism was not directly assessed. Many studies have assessed muscle protein metabolism in detail using intrusive methods, assessing nitrogen balance, amino acid oxidation indices, and responses to muscle biopsies [58,59,60]. Conversely, in this study, muscle protein metabolism was inferred through serum changes (changes in myokines and inflammation factors) and muscle strength measurements. Future studies are required to analyze detailed changes in muscle protein metabolism, based on positive changes in myokine levels, inflammation factors, and muscle quality after EAA + RE treatment, as revealed in our study.

## 5. Conclusions

The conclusions demonstrated in this study are shown in Figure 3.

This study confirmed the effects of RE, EAA, and RE + EAA over four weeks on body composition (lean body fat and lean body mass), blood myokine (irisin and myostatin A) and inflammation (TNF-α) factors, and muscle quality (knee–abdominal muscle strength and grip strength). The results revealed that RE, EAA, and RE + EAA increased irisin levels and decreased myostatin A and TNF-α levels. In addition, RE + EAA increased muscle quality (grip strength—left, trunk flexion peak torque, and keen flexion peak torque—left). Our results confirmed that EAA could improve myokine and inflammation factors similarly to RE. However, body fat mass was only significantly decreased after four weeks of RE or RE + EAA, and muscle mass was only significantly increased by RE + EAA. Finally, RE must be included in treatment to improve muscle quality over four weeks. The results of this study demonstrate that EAA intake could improve myokine and inflammation factor levels like RE. Four weeks of RE + EAA improved body fat, muscle mass, and myokine and inflammation factor levels the most markedly among all treatments. Therefore, the combined treatment of resistance exercise and EAA intake can be recommended to maintain the health of adults because it has a synergistic effect in improving muscle synthesis, myokine, and inflammation factors.

## Figures and Tables

**Figure 1 nutrients-16-01688-f001:**
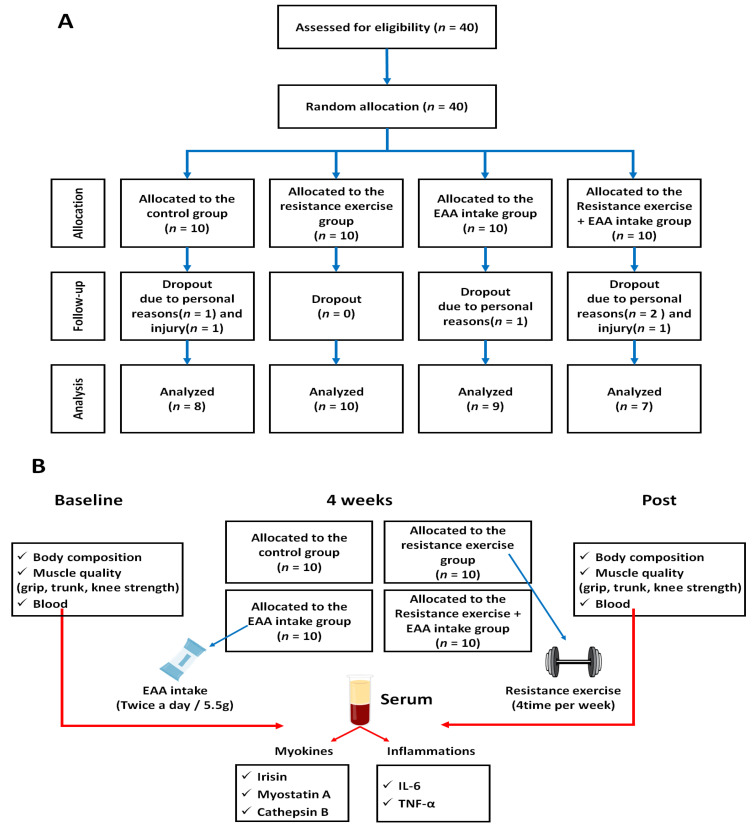
Study design. (**A**) Study flow diagram. (**B**) Experimental protocol.

**Figure 2 nutrients-16-01688-f002:**
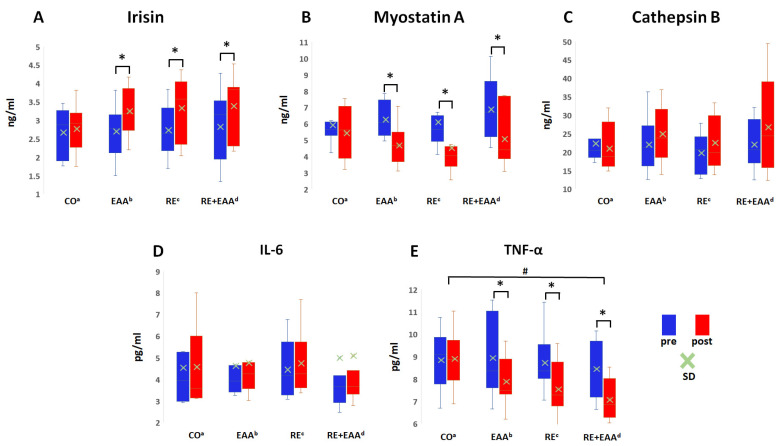
Changes in myokine and inflammation factors. (**A**) Irisin. (**B**) Myostatin A. (**C**) Cathepsin B. (**D**) IL-6. (**E**) TNF-α. Values are presented as mean ± SD. ^a^: CO, control group; ^b^: RE, resistance exercise group; ^c^: EAA, essential amino acids intake group; ^d^: RE + EAA, resistance exercise + essential amino acids intake group. * *p* < 0.05, within the group; # *p* < 0.05, interaction effect.

**Figure 3 nutrients-16-01688-f003:**
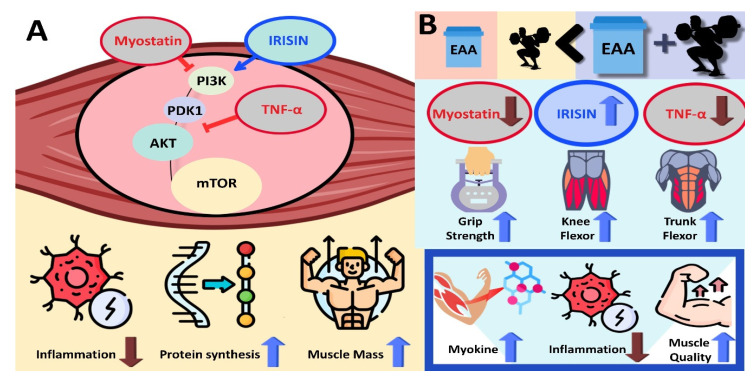
(**A**) Following regular resistance exercise and EAA treatment, a decrease in myostatin and TNF-α and an increase in IRISIN activate the mTOR signaling pathway → increasing muscle mass through a decrease in inflammation and the activation of protein synthesis. (**B**) Compared to EAA intake or resistance exercise alone, EAA intake and resistance exercise appear to be more positive for improving muscle quality and myokines. Therefore, a combined treatment of resistance exercise and EAA can be recommended to maintain health in adults because it has a synergistic effect in improving muscle synthesis, myokines, and inflammation factors.

**Table 1 nutrients-16-01688-t001:** Physical characteristics of the participants.

Treatment	Age	Height (cm)	Weight (kg)	BMI (kg/m^2^)	Lean Body Fat (kg)	Lean Body Mass (kg)
CO ^a^ (*n* = 8)	20.63 ± 1.06	174.51 ± 4.73	65.46 ± 4.49	21.54 ± 1.85	8.16 ± 2.44	32.65 ± 1.85
RE ^b^ (*n* = 10)	21.80 ± 1.03	173.98 ± 4.07	71.66 ± 7.04	23.69 ± 2.12	10.95 ± 2.68	34.48 ± 3.46
EAA ^c^ (*n* = 9)	21.67 ± 2.29	176.76 ± 4.01	67.20 ± 6.79	21.53 ± 2.20	11.04 ± 4.46	31.90 ± 1.81
RE + EAA ^d^ (*n* = 7)	22.57 ± 2.37	174.09 ± 3.56	75.69 ± 4.10	25.01 ± 1.49	12.41 ± 2.24	36.43 ± 1.93

Values are shown as mean BMI, Body Mass Index, ± SD, ^a^: CO, control group; ^b^: RE, resistance exercise group; ^c^: EAA, essential amino acids intake group; ^d^: RE +EAA, resistance exercise + essential amino acids intake group.

**Table 2 nutrients-16-01688-t002:** Components of the EAAs.

Variable	Total: 5500 mg
Sodium	40 mg
Carbohydrate	1000 mg
Fat	100 mg
L-Leucine	1389 mg
L-Valine	357 mg
L-Arginine	321 mg
L-Phenylalanine	218 mg
L-Histidine	53 mg
L-Isoleucine	347 mg
L-Threonine	304 mg
L-Methionine	107 mg
L-Lysine Monohydrochloride	562 mg
L-Tryptophan	2 mg
Flavor	700 mg

**Table 3 nutrients-16-01688-t003:** Resistance exercise program.

Composition	Exercise Program				Set	Time (min)	Intensity and Volume
Warm Up	Dynamic Stretching				1	10	
Resistance exercise	Mon, Thu	Part	Tue, Fri	Part	3	50	1~2 weeks 1RM/ 60~70% 3~4 weeks 1RM/ 70~80% 300~350 kcl 4 times Per week
	Bench Press	Chest	Squat	Leg			
			Leg Extension				
	Dumbbell Press						
			Leg Curl				
	Chest Dips						
			Leg Press				
	Cable Crossover Dead Lift	Back					
			Calf Raise				
	Lat Pull-down						
			Military Press	Shoulder			
	Seated Row						
			Side Lateral raise				
	Assisted Pull-up						
			Biceps Curl	Arm			
	Back Extension	Abdominals					
			Lying triceps Extension				
	Crunch						
			Crunch	Abdominals			
Cool down	Dynamic stretching				1	10	

**Table 4 nutrients-16-01688-t004:** Change in body composition of the participants.

Variable	Treatment	Baseline	4 Weeks	Group	F-Value (*p*-Value)
Height (cm)	CO ^a^	174.51 ± 4.73	174.51 ± 4.62	Baseline - 4 weeks -	G: 0.806 (*p* = 0.500) T: 1.14 (*p* = 0.294) G × T: 1.015 (*p* = 0.400)
	RE ^b^	173.98 ± 4.07	174.17 ± 4.11		
	EAA ^c^	176.76 ± 4.01	176.67 ± 3.96		
	RE + EAA ^d^	174.09 ± 3.56	174.30 ± 3.71		
Weight (kg)	CO ^a^	65.46 ± 4.49	65.51 ± 5.01	Baseline d > a, c c > a 4 weeks d > a, c b > a	G: 4.625 (*p* = 0.009) T: 3.272 (*p* = 0.081) G × T: 0.668 (*p* = 0.578)
	RE ^b^	71.66 ± 7.04	71.04 ± 6.25		
	EAA ^c^	67.20 ± 6.79	66.93 ± 6.10		
	RE + EAA ^d^	75.69 ± 4.10	74.86 ± 3.77		
BMI (kg/m^2^)	CO ^a^	21.54 ± 1.85	21.55 ± 1.85	Baseline d, c > a, b 4 weeks d, c > a, b	G: 5.976 (*p* = 0.003) T: 4.939 (*p* = 0.034) G × T: 1.146 (*p* = 0.347)
	RE ^b^	23.69 ± 2.12	23.41 ± 1.77		
	EAAs ^c^	21.53 ± 2.20	21.46 ± 2.04		
	RE + EAA ^d^	25.01 ± 1.49	24.64 ± 1.28		
Lean body fat (kg)	CO ^a^	8.16 ± 2.44	8.56 ± 2.83	Baseline - 4 weeks -	G: 1.925 (*p* = 0.147) T: 7.893 (*p* = 0.009) G × T: 3.585 (*p* = 0.025)
	RE ^b^	10.95 ± 2.68	10.48 ± 2.34 *		
	EAA ^c^	11.04 ± 4.46	10.33 ± 3.88		
	RE + EAA ^d^	12.41 ± 2.24	11.60 ± 2.58 *		
Lean body mass (kg)	CO ^a^	32.65 ± 1.85	32.44 ± 2.08	Baseline d > a 4 weeks d > b > a, c	G: 7.006 (*p* = 0.001) T: 6.136 (*p* = 0.019) G × T: 4.513 (*p* = 0.010)
	RE ^b^	34.48 ± 3.46	34.53 ± 3.17		
	EAA ^c^	31.90 ± 1.81	32.21 ± 1.63		
	RE + EAA ^d^	36.43 ± 1.93	37.49 ± 1.38 *		

Values are mean BMI, Body Mass Index, ± SD, ^a^: CO, control group; ^b^: RE, resistance exercise group; ^c^: EAA, essential amino acids intake group; ^d^: RE +EAA, resistance exercise + essential amino acids intake group. * *p* < 0.05 difference within the group.

**Table 5 nutrients-16-01688-t005:** Change in muscle quality (grip, trunk, and knee strength).

Variable	Treatment	Baseline	4 Weeks	Group	F-Value (*p*-Value)
Grip strength (L) (kg)	CO ^a^	41.63 ± 3.14	43.69 ± 4.43	Baseline d > a, c 4 weeks d > a, c	G: 3.550 (*p* = 0.028) T: 4.457 (*p* = 0.044) G × T: 0.203 (*p* = 0.984)
	RE ^b^	44.49 ± 4.51	45.70 ± 5.57		
	EAA ^c^	41.50 ±6.83	42.36 ± 3.03		
	RE + EAA ^d^	47.77 ± 3.03	48.71 ± 3.02 *		
Grip strength (R) (kg)	CO ^a^	44.51 ± 2.35	45.75 ± 3.13	Baseline - 4 weeks -	G: 1.776 (*p* = 0.174) T: 10.748 (*p* = 0.003) G × T: 0.470 (*p* = 0.706)
	RE ^b^	46.37 ± 2.58	49.30 ± 4.38 *		
	EAA ^c^	45.44 ± 4.52	48.00 ± 5.13		
	RE + EAA ^d^	48.97 ± 6.39	50.40 ± 5.39		
Trunk extension peak torque [(Newton-meter/body weight (kg)]	CO ^a^	321.25 ± 51.09	315.00 ± 35.44	Baseline b > a, b, c 4 weeks a, b, d > c	G: 3.241 (*p* = 0.036) T: 0.602 (*p* = 0.444) G × T: 0.868 (*p* = 0.468)
	RE ^b^	329.20 ± 58.64	343.00 ± 30.16		
	EAA ^c^	287.11 ± 39.72	280.67 ± 41.99		
	RE + EAA ^d^	300.14 ± 26.84	323.00 ± 48.30		
Trunk flexion peak torque [(Newton-meter/body weight (kg)]	CO ^a^	289.25 ± 16.11	288.63 ± 27.94	Baseline - 4 weeks -	G: 1.161 (*p* = 0.341) T: 4.174 (*p* = 0.050) G × T: 2.962 (*p* = 0.048)
	RE ^b^	286.20 ± 25.18	294.90 ± 24.62 *		
	EAA ^c^	276.56 ± 19.01	273.33 ± 12.63		
	RE + EAA ^d^	279.71 ± 15.03	293.00 ± 17.45 *		
Knee extension peak torque (R) [Newton-meter/body weight (kg)]	CO ^a^	280.87 ± 50.43	275.25 ± 55.91	Baseline - 4 weeks -	G: 2.320 (*p* = 0.095) T: 0.882 (*p* = 0.355) G × T: 1.741 (*p* = 0.180)
	RE ^b^	306.70 ± 44.26	309.40 ± 45.17		
	EAA ^c^	261.44 ± 43.68	259.78 ± 22.74		
	RE + EAA ^d^	287.29 ± 35.53	306.43 ± 37.06		
Knee extension peak torque (L) [Newton-meter/body weight (kg)]	CO ^a^	271.13 ± 46.51	268.12 ± 48.46	Baseline - 4 weeks -	G: 1.700 (*p* = 0.188) T: 0.149 (*p* = 0.702) G × T: 1.419 (*p* = 0.257)
	RE ^b^	290.30 ± 40.71	308.00 ± 38.62		
	EAA ^c^	269.11 ± 47.98	257.89 ± 42.07		
	RE + EAA ^d^	293.14 ± 39.99	298.14 ± 47.65		
Knee flexion peak torque (R) [(Newton-meter/body weight (kg)]	CO ^a^	140.75 ± 19.29	149.75 ± 22.89	Baseline - 4 weeks -	G: 2.627 (*p* = 0.068) T: 2.867 (*p* = 0.101) G × T: 0.686 (*p* = 0.568)
	RE ^b^	158.90 ± 10.21	163.70 ± 11.38		
	EAA ^c^	143.00 ± 18.13	141.78 ± 18.89		
	RE + EAA ^d^	147.29 ± 19.81	152.14 ± 19.00		
Knee flexion peak torque (L) [(Newton-meter/body weight (kg)]	CO ^a^	139.37 ± 27.24	139.75 ± 27.57	Baseline - 4 weeks -	G: 1.714 (*p* = 0.185) T: 4.618 (*p* = 0.040) G × T: 0.614 (*p* = 0.611)
	RE ^b^	151.10 ± 24.25	158.00 ± 20.98		
	EAA ^c^	134.44 ± 18.48	139.00 ± 12.77		
	RE + EAA ^d^	146.14 ± 17.99	156.71 ± 15.59 *		

Values are presented as mean ± SD. ^a^: CO, control group; ^b^: RE, resistance exercise group; ^c^: EAA, essential amino acids intake group; ^d^: RE +EAA, resistance exercise + essential amino acids intake group. * *p* < 0.05 within the group.

## Data Availability

Data is contained within the article. The data presented in this study are available on request from the corresponding author.

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
