# Peer review of "Effects of Resistance Exercise and Essential Amino Acid Intake on Muscle Quality, Myokine, and Inflammation Factors in Young Adult Males"

_nutrients, 2024, doi:10.3390/nu16111688_

Round 1

Reviewer 1 Report

Comments and Suggestions for Authors

Jeong_essential amino acid_nutrients_2024

I commend the authors on the completion of this manuscript. I really liked the theme and design. Overall it is well designed and on an innovative and important topic. I have a few concerns highlighted below.

Abstract

Line 31. Please, review: “(ANOVA) revealed a significant interaction effect on lean body fat (p < 0.05) and lean body (p < 0.05)”

1. Introduction

The introduction and the aim of the study do not speak about inflammation factors, but they are measured and analyzed. Please correct this issue. 

2. Materials and Methods

Line 101. Sample size was calculated using G* power software version 3.1.9.7. Could you please added the data used for the calculation of the sample size? Effect size or means and SD? 

Line 109. During the experiment period, excluding weekends, average dietary calories were 2738 ± 124 Kcal (carbohydrate: 343 ± 65 Kcal; protein: 123 ± 29 Kcal; fat: 87 ± 19 Kcal). Please added: average daily dietary calories. When you are speaking about carbohydrate, protein and fat I think that you are speaking about grams, not calories. 

Line 119. Lean mass, please correct to lean body mass in all the text and tables of the manuscript.

Table 2. Please avoid the use of etc and write the total composition. 

2.7. Statistical analysis

Have you checked the assumptions of the models? Box test or similar. Why have you used such a complicated statistics with a very limited sample size? I think that non-parametric statistics, with Kruskal-Wallis (post-hoc also) and Wilcoxon test might have been better. 

3. Results

Line 190. Weight showed a tendency of the main effect (p = 0.081). Please add of time. 

4. Discussion

Line 259. Please, correct. This sentence is not understandable: “Thus, it was a short-term (5 weeks). Whether RE and EAA treatment was effective in increasing 260

muscle mass was unclear.”

Line 273: “Thus, it is necessary to confirm the most effective time of RE+EAA considering the subject's exercise intensity, gender, and age.” Please add some sentence and reference in the previous paragraph speaking about differences according to age. 

Line 289. “Our study also found significant changes in left grip (p < 0.05), trunk flexion peak torque (p < 0.05), and knee flexion peak torque R (p < 0.05) in proportion to the increase in muscle mass in the RE + EAA group.” You have not correlated the changes in muscle mass with the changes in strength, thus, you have not to use the term in proportion in the previous sentence. 

Line 292. “Interestingly, despite a short-term treatment (RE + EAA) for 4 weeks, the adaptive response of muscle nerves [41],… synthesis rate [35]”. Please syntax review. 

Line 310. Please, add, “inflammation biomarkers” to the title. 

Line 348. Please review the next sentence and remove irisin from it. “Therefore, RE and EAA intake are effective in inhibiting the activation of irisin and myostatin A, thereby suppressing the expression of mTOR and inflammation cytokines.”

Conclusions

Line 372: “Results revealed that RE, EAA, and RE+EAA increased irisin and myostatin A levels but decreased TNF-α levels.” Please review Myostatin was not increased. 

Author Response

Dear Nutrient’s editors and reviewers,

Thank you very much for taking the time to review this manuscript.

Our study focused on whether essential amino acids (EAA) supplements mimic resistance exercise (RE) and whether there is a synergistic effect of RE with EAA supplements. Generally, RE and EAA are well known for muscle synthesis, and we know for molecular mechanisms related to protein synthesis. However, except for, we do not know well how other factors change and are affected or how quickly the effect takes effect. It means that if other factors related to muscle (myokine) positively change through EAA or RE combined with EAA, we may be able to access for solving the various diseases. We added the answer to your review.

Reviewer 2 Report

Comments and Suggestions for Authors

Thank you for the opportunity to review your manuscript, “Resistance exercise and essential amino acid intake would affect muscle quality, myokine and inflammation factors in young adult males?”

The study aims to investigate changes in muscle function, myokines and biomarkers of inflammation as a function of endurance physical activity and (or) EAA supplementation.

The abstract exceeds the 200 words required by the journal. It should be reworded as soon as possible.

Line 48. This statement is not consistent with the article presented before COVID-19.Its rate has been gradually increasing since COVID-19

Figure 1 is not cited in the text.

Material and methods should start with the type of study design.

Line 101-103. It is mentioned that a calculation was made, but not the parameters used and from which article the necessary statistics were extracted. This makes the calculation irreproducible.

It is mentioned that 34 people were needed, but we need to know if this is the total or per group. They were divided into four groups, losing six subjects and three from one group. This results in very small samples, which detracts from the study's external validity..

Providing readers with a picture of how the evaluations were carried out can be instructive. I recommend that authors consider this possibility.

Line 224-229. I recommend the authors give the actual value and not the approximation.

If there is no group-time interaction, there is no point in doing it, and it may create the impression that there is an effect when there is not.

It is a limitation of the study not to start from the same level of training as the subjects. This factor should be reflected in the limitations.

The conclusions of the study should not be a graph.

Line 370-272. It should specify what effects are produced.

Given the study's small sample size and statistical results, the conclusions are optimistic.

Author Response

(The authors gave the same response as above.)

Round 2

Reviewer 1 Report

Comments and Suggestions for Authors

I thank the authors the correction of the manuscript according to our suggestions. 

Reviewer 2 Report

Comments and Suggestions for Authors

The authors have responded to my queries and made the necessary corrections.